# OptSHAP: Explaining Dimensionality Reduction-based Models for Tabular Data via Optimization

## Abstract

Dimensionality Reduction-based Models (DRbMs), which couple a dimensionality reduction technique with a predictive model, are commonly used to mitigate overfitting and reduce computational complexity regarding high-dimensional tabular data. However, their two-stage architecture presents considerable challenges for explainability, as the projection obscures the original feature space, thus making the model output difficult to interpret in terms of the input features. Model-agnostic explanation methods are applicable to DRbMs but typically rely on sampling-based approximations, leading to instability and low-faithfulness explanations. To address these limitations, we introduce OptSHAP, the first optimization-based attribution specifically designed for DRbMs. Our method leverages reduced-space attributions and then redistributes them back to the original feature space through a transformation that satisfies the principle of efficiency. Additionally, we propose a novel evaluation metric, the $k$-Local Stability Score (LSS), which quantifies the stability of feature attribution methods by averaging their distances to local explanations. Extensive empirical evaluations across high-dimensional datasets, various dimensionality reduction techniques, and multiple machine learning models demonstrate that OptSHAP outperforms state-of-the-art attribution methods, achieving up to $24\times$ stability and $2\times$ fidelity on key benchmarks.

## 1 Introduction

Learning from high-dimensional tabular datasets has become increasingly common across various domains (Ruiz et al., 2023). Such high-dimensional feature spaces give rise to the well-known "curse of dimensionality", which increases computational cost and overfitting risk (Köppen, 2000). A popular and effective strategy to mitigate these issues is to use a Dimensionality Reduction-based Model (DRbM), where meaningful features are first extracted through a dimensionality reduction technique and then passed to a machine learning model. Specifically, Dimensionality Reduction (DR) techniques, such as Principal Component Analysis (PCA (Abdi & Williams, 2010)), Independent Component Analysis (ICA (Hyvärinen et al., 2009)) and Linear Discriminant Analysis (LDA (Balakrishnama & Ganapathiraju, 1998)), project the data into a lower-dimensional space (called a reduced space), thereby reducing computational cost, and facilitating data visualization (Jia et al., 2022; Velliangiri et al., 2019). Moreover, by focusing on the most informative components, DR techniques may help improve model accuracy on downstream tasks (Hasan & Abdulazeez, 2021). For example, DRbMs are practically effective in clinical prediction tasks, genomics, and bioinformatics, where tabular datasets often contain thousands of features (Fu & Wang, 2003; Singh et al., 2016). Similarly, Hollmann et al. (2025) point out that DR techniques may be beneficial for supervised tabular learning models designed for small- to moderate-sized datasets.

While their empirical performance is well established, DRbMs, like many machine learning models, face a trade-off between predictive performance and explainability (Bell et al., 2022). This trade-off becomes especially pronounced in DRbMs due to their two-stage structure, which complicates the task of attributing model predictions to original input features. Thus, explaining DRbMs becomes an essential task, both for model debugging and for building trust in practical applications. Due to DRbM's complexity, post-hoc explanations have become a common paradigm, where the focus lies in local feature attributions, explaining why the model makes a specific prediction for a single

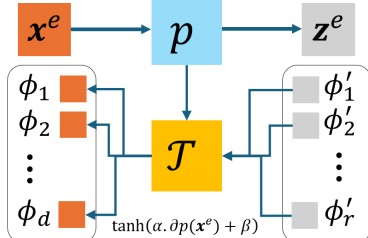

Figure 1: General (**Left**) and Internal (**Right**) view of the OptSHAP framework. These show how reduced explanations are reallocated back to the original feature space via a transformation block $\mathcal{T}$, which integrates gradient information $J_p(\boldsymbol{x}^e)$ through a $\texttt{tanh}$ gate, where $\boldsymbol{x}^e$ is an explicand.

sample (known as *explicand* (Sundararajan & Najmi, 2020)). We categorize local feature attribution methods into model-agnostic and model-specific approaches. Existing *model-agnostic* methods (e.g., EIC (Strumbelj & Kononenko, 2010), KernelSHAP (Lundberg & Lee, 2017), and LIME (Ribeiro et al., 2016)) are applicable regardless of the underlying model. While they can be used to explain a DRbM, they suffer from two notable shortcomings: (1) their sampling-based estimates of feature importance are inherently unstable (Chen et al., 2022); and (2) they estimate the explanation directly through the pipeline, without considering how it behaves in the reduced space, which may result in lower confidence. On the other hand, *model-specific* methods, designed for particular model types, are typically much faster and meaningfully deeper than model-agnostic ones. Examples include: DeepLIFT (Shrikumar et al., 2017), DeepSHAP (Lundberg & Lee, 2017) and Integrated Gradients (Sundararajan et al., 2017) are for deep models, TreeSHAP (Lundberg et al., 2020) is for tree-based models. Yet these methods are not capable of explaining a series of models (Chen et al., 2022), including DRbMs.

Therefore, there is a pressing need for a more effective feature attribution approach specialized for DRbMs, which can achieve two key objectives: ($\mathcal{O}1$) avoiding reliance on sampling, thereby reducing variance in the explanations; and ($\mathcal{O}2$) leveraging the explanation generated by model-specific methods in the reduced space.

**Contributions**. We introduce OptSHAP, a *novel* optimization-based attribution framework for explaining DRbMs. This maps the explanations from the reduced space back to the original feature space using a transformation block $\mathcal{T}$, which incorporates gradient information $J_p(\boldsymbol{x}^e)$ through a $\texttt{tanh}$ gating mechanism (see Fig. 1). *To the best of our knowledge, it is the first approach to exploit explanations in the reduced space and reproject them onto the original feature space*. The novelties and contributions are briefly as follows:

- First, we propose an optimization-based attribution, namely OptSHAP, which achieves two objectives ($\mathcal{O}1$) reduced-variance explanations and ($\mathcal{O}2$) model-specific alignment. We support this with theoretical analyses demonstrating that this approach satisfies desirable properties, including stability (c.f. Sec. 4).

- Second, we introduce the *k-Local Stability Score (LSS)*, a new metric for evaluating the stability of attribution methods. Compared to existing perturbation-based metrics such as Stability and Relative Stability, LSS is more computationally efficient, flexible, and easier to interpret (c.f. Sec. 5).

- Third, we conduct comprehensive experiments across high-dimensional tabular datasets using three DR techniques and four classifiers to assess the faithfulness of OptSHAP relative to three state-of-the-art (SOTA) attribution methods for DRbMs. Our results demonstrate improved performance in both quantitative metrics and qualitative evaluations (c.f. Sec. 6).

## 2 RELATED WORKS

**Explaining DRbMs via model-agnostic methods**. Model-agnostic (post-hoc) methods approximate feature attributions through perturbation or sampling, and thus can be applied to DRbMs (Slack et al., 2021).

One example of such methods, Explaining Individual Classification (EIC (Strumbelj & Kononenko, 2010)), estimates feature contributions by randomly masking subsets of features and measuring the change in the model's output when a feature is perturbed. While general, it suffers from high variance and requires many evaluations, becoming impractical for high-dimensional data (only tested up to 279 features in the original paper (Strumbelj & Kononenko, 2010)). Next, KernelSHAP (Lundberg & Lee, 2017) is widely used and theoretically grounded in Shapley values, offering a principled model-agnostic explanation framework. Yet, its independence assumptions rarely hold for DRbMs (Christoph, 2020), where the transformed data is a combination of all original features. In addition, reliable estimates in high dimensions demand many samples, making it computationally expensive (Lundberg & Lee, 2017). From a different approach, Local Interpretable Model-Agnostic Explanations (LIME (Ribeiro et al., 2016)) is attractive for its flexibility and low runtime, fitting simple surrogates around an explicand. However, explanations vary significantly across runs (Alvarez-Melis & Jaakkola, 2018), depend on ad-hoc choices of surrogate complexity, and can be adversarially manipulated to obscure bias (Slack et al., 2020), limiting trust in sensitive applications. Overall, while broadly applicable, model-agnostic methods struggle with instability and low-faithfulness in the DRbM context, motivating the need for more specialized approaches.

**Model-specific explanation methods**. These methods are tailored to particular architectures or tasks and are usually faster and more faithful than model-agnostic approaches (Viswan et al., 2024). However, they cannot be directly applied to DRbMs, since intermediate dimensionality reduction steps break their attribution rules. For example, TreeSHAP provides fast and exact Shapley values for tree models (Lundberg et al., 2020), while DeepSHAP (Lundberg & Lee, 2017) approximates Shapley values for neural networks by combining them with DeepLIFT (Shrikumar et al., 2017). Generalized DeepSHAP (G-DeepSHAP (Chen et al., 2022)) extends DeepSHAP with a rescale rule that aims to propagate attributions across heterogeneous model pipelines. However, this rule requires explanations from each layer, which fails for intermediate transformations such as dimensionality reduction, whose outputs are feature representations rather than predictions. Moreover, the rescale step can be numerically unstable: when explicand and baseline outputs are close, the denominator becomes small and leads to inflated attribution values (Appendix D). This distorts the explanation, making it sensitive to noise and difficult to interpret in practice. In summary, model-specific methods are efficient and faithful within their target architectures but cannot handle DRbMs, leaving a gap that motivates our proposed approach and connects to recent advances on pipeline explainability.

**Recent work on representation-level and pipeline explainability**. Recent studies also examine how feature transformations and pipeline design influence explanations. For example, Hwang et al. (2025) shows that binning or encoding features can shift SHAP rankings drastically. Similarly, the choice of transformation and model pair is demonstrated to alter what features appear important (Karwowska et al., 2025). In addition, Gwinner et al. (2024) introduce representation-level explanation similarity analysis as an additional pipeline stage, emphasizing that pipeline design directly shapes how explanations are produced. These works underline that representation and pipeline structure materially influence explanations, a gap we address with OptSHAP by reallocating attributions accounting for the dimensionality reduction transform.

## 3   PROBLEM FORMULATION

We begin with a toy example shown in Fig. 2. Three features (age, height, cholesterol) are projected by PCA into two components $PC_1$ and $PC_2$. A model $f$ operating on these components produces an output of 0.9, with attributions $\phi'_1 = \phi_{PC_1} = 0.6$ and $\phi'_2 = \phi_{PC_2} = 0.3$. In practice, $PC_1$ and $PC_2$ are abstract combinations of the original variables. They do not admit a direct semantic interpretation in the same way as the input features (i.e, age, height, or cholesterol).

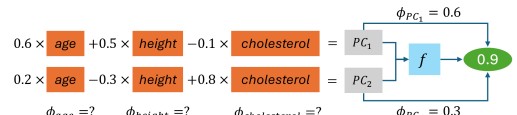

Figure 2: Toy example illustrating the challenge: PCA components are not directly interpretable, requiring reallocation of importance scores back to original features.

The challenge, therefore, is to reassign their importance scores to the original feature space. This reallocation must satisfy two properties: (i) *sign preservation* and (ii) *efficiency*. For instance, in $PC_1$ the loadings of height (0.5) and cholesterol ($-0.1$) have opposite signs; thus, a positive attribution $\phi'_1$ yields a positive allocation to height and a negative one to cholesterol. Moreover, an efficiency redistribution could be 0.39 to age, 0.28 to height, and $-0.08$ to cholesterol, which sums close to

$\phi'_1$, allowing for a small approximation error due to allocation constraints. Generalizing from this toy case, we now formalize the problem for a general DRbM, where attributions derived from the reduced space must be redistributed back to the original feature space.

**Problem statement**. Consider a DRbM consisting of a projection $p$ followed by a model $f$: $\mathcal{X} \xrightarrow{\text{projection } p} \mathcal{Z} \xrightarrow{\text{model } f} \mathcal{Y}$, where $p : \mathbb{R}^d \to \mathbb{R}^r$ is a linear DR technique. Given an explican $\boldsymbol{x}^e = (x_1, \ldots, x_d)$, the prediction is $y^e = (f \circ p)(\boldsymbol{x}^e) = f(\boldsymbol{z}^e)$ with $\boldsymbol{z}^e = p(\boldsymbol{x}^e)$. Moreover, each reduced component $\mathbf{z}_i$ can be expressed through a component function $p_i$ as $\mathbf{z}_i = p_i(\mathbf{x}_1, \mathbf{x}_2, \ldots, \mathbf{x}_d)$.

Our task is to provide the explanation for $y^e$ in the original space, $\Phi(f \circ p, \boldsymbol{x}^e) = (\phi_1, \phi_2, \ldots, \phi_d)$, by propagating the reduced-space explanation $\Phi(f, \boldsymbol{z}^e) = (\phi'_1, \phi'_2, \ldots, \phi'_r)$ through the component functions $p_1, p_2, \ldots, p_r$. The idea is to reallocate each $\phi'_i$ to the input features, formalized as a transformation $\mathcal{T}$ with allocation terms $\phi^{\mathcal{T}}_{i \to j}$. Inspired by ideas from cooperative game theory, particularly the efficiency property of Shapley value (Lundberg & Lee, 2017), $\mathcal{T}$ is required to satisfy the two following properties to ensure meaningful allocation:

- (Property 1) *Sign preservation*:

$$\phi'_i . \partial^i_j . \phi^{\mathcal{T}}_{i \to j} > 0, \text{ for all } i, j, \tag{1}$$

where $\partial^i_j := \frac{\partial p_i}{\partial x_j}\big|_{\boldsymbol{x}^e}$ arre the partial derivatives of $p_i$ with respect to $\mathbf{x}_j$ at $\boldsymbol{x}^e$, i.e., entries of the projection Jacobian $J_p(\boldsymbol{x}^e)$. This assures the interaction of $\phi_i$ and $\phi^{\mathcal{T}}_{i \to j}$ will be positive if $\partial^i_j > 0$ and negative if $\partial^i_j < 0$.

- (Property 2) *Efficiency*:

$$\sum_{j=1}^{d} \phi_j = \sum_{j=1}^{d} \sum_{i=1}^{r} \phi^{\mathcal{T}}_{i \to j} \approx \sum_{i=1}^{r} \phi'_i, \tag{2}$$

meaning that the total allocated contributions should be close to the total original importance. This relaxed view of efficiency has been noted in prior work, including accepting up to a 5% deviation (Sundararajan et al., 2017), the WeightedSHAP method that explicitly relaxes the axiom (Kwon & Zou, 2022), and stochastic approaches like SIM-Shapley allowing low-bias deviations for efficiency (Li et al., 2025).

## 4 THE PROPOSED FRAMEWORK: OPTSHAP

To illustrate the idea of OptSHAP, let us revisit the toy example. Tab. 1 shows how the attributions are redistributed: for PC$_1$, OptSHAP assigns positive contributions to age (0.39) and height (0.28), while a negative contribution is assigned to cholesterol ($-0.08$) due to its negative loading. A similar redistribution should be observed for PC$_2$, resulting in feature-level scores $(0.49, 0.16, 0.24)$ that are more interpretable than component-level attributions. To formalize this idea, we introduce in Fig. 1 a transformation block $\mathcal{T}$, which combines information from $p$ and $\Phi(f, \boldsymbol{z}^e)$ to reallocate reduced-space contributions $\phi'_i$ to the original feature space $(\phi_1, \ldots, \phi_d)$.

Table 1: OptSHAP reallocation from reduced to original features on the toy example. The "Loadings" rows show PCA loadings for each component (at $\mathbf{x}^e$); the "Alloc." rows are the resulting allocations $\phi^{\mathcal{T}}_{i \to j}$.

| | | $\phi_{\text{age}}$ | $\phi_{\text{height}}$ | $\phi_{\text{chol.}}$ | Sum |
|---|---|---|---|---|---|
| $\phi'_1$ | Alloc. | **0.39** | **0.28** | **$-0.08$** | 0.59 |
| | Loadings | 0.6 | 0.5 | $-0.1$ | – |
| $\phi'_2$ | Alloc. | **0.10** | **$-0.12$** | **0.32** | 0.30 |
| | Loadings | 0.2 | $-0.3$ | 0.8 | – |
| | Total | 0.49 | 0.16 | 0.24 | 0.89 |

Now, if $\mathcal{T}$ allocates $\phi'_i$ to $\phi_j$ with contribution $\phi^{\mathcal{T}}_{i \to j}$, the residual of this process is:

$$\mathcal{R}_i = \phi'_i - \sum_{j=1}^{d} \phi^{\mathcal{T}}_{i \to j}. \tag{3}$$

The transformation $\mathcal{T}$ is then designed to minimize $\mathcal{R}_i$ while ensuring the two aforementioned redistribution properties. To this end, we utilize the parametric hyperbolic tangent function defined below.

---

**Algorithm 1** OptSHAP: Explaining a DRbM via optimization

---

**Input:** DRbM with data $\mathcal{X}^d$, DR technique $p$, model $f$, explicand $\boldsymbol{x}^e$.
**Output:** $\Phi(f \circ p, \boldsymbol{x}^e) = (\phi_1, \ldots, \phi_d)$.
**Procedure:**
1: Compute reduced explanation $\Phi(f, p(\boldsymbol{x}^e)) = (\phi_1', \ldots, \phi_r')$.
2: Solve optimization problem (5) for $\alpha_1, \ldots, \alpha_r, \beta$.
3: **for** $j = 1$ to $d$ **do**
4:     $\phi_j = \sum_{i=1}^r \phi_i' \tanh(\alpha_i \partial_j^i + \beta)$.
5: **end for**
6: **return** $(\phi_1, \ldots, \phi_d)$.

---

**Definition 4.1** (Allocation Transform). Given parameters $\{\alpha_i > 0\}_{i=1}^r$ and a shift $\beta \in \mathbb{R}$, the transformation $\mathcal{T}$ reallocates $\phi_i'$ to feature $x_j$ as $\phi_{i \to j}^{\mathcal{T}} = \tanh(\alpha_i \partial_j^i + \beta) \phi_i'$.

Consequently, the residual can be expressed as:

$$\mathcal{R}_i = \phi_i'(1 - \sum_{j=1}^d \tanh(\alpha_i \cdot \partial_j^i + \beta)). \tag{4}$$

Here, the $\tanh(.)$ is employed because it is bounded, monotonic, and symmetric around zero. These properties avoid exploding attributions and ensure stable redistribution, whereas other common activation functions, such as softmax and sigmoid, fail to meet the symmetry and sign-preservation requirement since their outputs are always non-negative. The sensitivity analysis for $\tanh(.)$ parameters is detailed in Appendix A.

The following theorem shows that $\mathcal{T}$ preserves the sign of contributions so that the reduced components maintain their directional relationships with the original features, see the proof in Appendix A.

**Theorem 4.2** (Sign Preservation). *Transformation $\mathcal{T}$ satisfies the property 1 about sign preservation.*

To ensure that the contribution of each original feature is recovered, we set $\phi_j = \sum_{i=1}^r \phi_{i \to j}^{\mathcal{T}}$. The property 2 is satisfied by minimizing the residual in Eq. 4. This leads to the following optimization problem:

$$\min_{\alpha_1, \alpha_2, \ldots, \alpha_r, \beta} \quad \sum_{i=1}^r \mathcal{R}_i^2. \tag{5}$$

We now describe the entire procedure in Alg. 1 named OptSHAP, which addresses the stated explainability challenges in DRbMs.

*Remark* 4.3. In this work, we consider $f$ as a tree-based model because tree-based models have been shown to outperform deep learning-based models on tabular data (Grinsztajn et al., 2022). Accordingly, an exact model-specific method for such models called TreeSHAP (Lundberg et al., 2020) is leveraged to gain the explanations in the reduced space. In this way, the propagation of attributions to the original space is more accurate (Chen et al., 2022).

In practice, gradient perturbations may arise when derivatives are computed approximately (e.g., via finite differences or stochastic estimation) or are affected by floating-point errors in high-dimensional projections. To account for such effects, we provide Theorem 4.4, which establishes an upper bound on the expected deviation of the allocation under Gaussian noise.

**Theorem 4.4** (Stability). *If the gradient information $J_p(\boldsymbol{x}^e)$ is perturbed by independent Gaussian noise $\delta_{ij} \sim \mathcal{N}(0, \sigma^2)$, then the expected deviation of the allocated importance $\phi_j$ satisfies:*

$$\mathbb{E}\left[|\phi_j - \phi_j^{noise}|\right] \leq \sum_{i=1}^r |\phi_i'| \, \alpha_i \min\left(2, \sigma\sqrt{\frac{2}{\pi}}\right). \tag{6}$$

## 5 FAITHFULNESS AND STABILITY EVALUATION

To evaluate attribution-based explanation methods, we adopt established benchmark metrics for faithfulness and stability as follows:

For faithfulness, we use MoRF, LeRF, and ABPC (Li et al., 2023). These metrics measure the effect of removing features ranked by attribution importance. MoRF removes the most relevant features first, showing sensitivity to critical inputs; LeRF removes the least relevant first, reflecting robustness to discarding uninformative inputs; and ABPC (area between perturbation curves) quantifies the separation between the two. Since ABPC already summarizes the MoRF and LeRF relationship, we omit LeRF for brevity.

For stability, perturbation-based metrics such as Infidelity (INFD (Yeh et al., 2019)), Stability (Alvarez-Melis & Jaakkola, 2018), and Relative Stability (Agarwal et al., 2022), provided that an effective sampling strategy is employed to estimate the empirical expectation or supremum in their formulation. However, these metrics may incur high computational complexity in large input spaces due to the need for repeated perturbations and model predictions across sampled inputs. Therefore, we propose a new metric, $k-$Local Stability Score ($k-$LSS), to quantify the model stability by leveraging the correlation of available explanations for explicands, thereby requiring low computational complexity. In addition, this metric is flexible, as it captures variation in explanations by adjusting the number of neighboring samples considered. Formally, given a black-box function $g$, an explanation functional $\Phi$, the formulation for the $k-$LSS as follows:

$$LSS(\boldsymbol{x}) = \frac{1}{k} \sum_{\boldsymbol{x}' \in \mathcal{N}_k(\boldsymbol{x})} \|\Phi(g, \boldsymbol{x}) - \Phi(g, \boldsymbol{x}')\|_2, \tag{7}$$

where $\mathcal{N}_k(\boldsymbol{x})$ denotes the set of $k$ nearest neighbors of the sample $\boldsymbol{x}$ with respect to the Euclidean ($\ell_2$) distance. Stability, in this context, refers to the principle that similar inputs should yield similar explanations (Agarwal et al., 2022). Thus, lower LSS values indicate higher stability.

In summary, we evaluate the faithfulness of attribution-based methods using four benchmarks: MoRF, ABPC, INFD, and LSS. Formal definitions are provided in Appendix C.

# 6 EXPERIMENTS

## 6.1 EXPERIMENTAL SETUP

**Baselines**. We compare OptSHAP with three SOTA methods (KernelSHAP, EIC, and LIME) in generating explanations for DRbMs based on the evaluation guideline in Sec. 5. High-quality explanations must be faithful, stable, and reliable, independent of the underlying model's predictive performance. Hence, our primary focus lies in evaluating the effectiveness of attribution-based explanation methods rather than comparing model accuracies. Nevertheless, we also demonstrate that DR techniques, such as PCA, ICA, and LDA, not only accelerate training compared to standalone XGBoost, but also enhance its classification accuracy, as illustrated in Fig. 3.

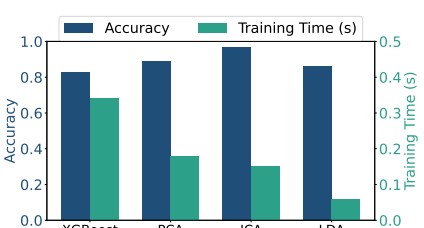

Figure 3: Accuracy and Training Time of XGBoost and DRbMs on the Alzheimer's Dataset.

**Datasets**. We consider three datasets for experiments, including Alzheimer's (Cilia et al., 2018), Toxicity (Gul et al., 2021), and Alon (Alon et al., 1999). All are high-dimensional datasets ranging from 451 to 2001 dimensions.

**Models**. We employ three linear projections: PCA, ICA, and LDA; and four tree-based models: XGBoost, CatBoost, LightGBM, and Random Forest.

**Evaluation metrics**. We utilize four metrics for faithfulness evaluation, which are described in Sec. 5, including MoRF and ABPC (higher is better); LSS, and INFD (lower is better).

Further details about the experiment setting are in Appendix B. We commit to publishing the implementation code if the paper is accepted.

## 6.2 OPTSHAP THROUGH SIMULATION

This section provides a simple simulation to show how OptSHAP works in Fig. 4. Specifically, we construct a 10-dimensional input that is linearly projected into a 4-dimensional reduced space. TreeSHAP first produces scores $(\phi'_1, \phi'_2, \phi'_3, \phi'_4)$ in the reduced space. OptSHAP maps these values back to the original 10 features while satisfying the *sign preservation* and *efficiency* properties. The left panel shows how the local attribution score $\phi'_2$ from the reduced space is redistributed for the 10

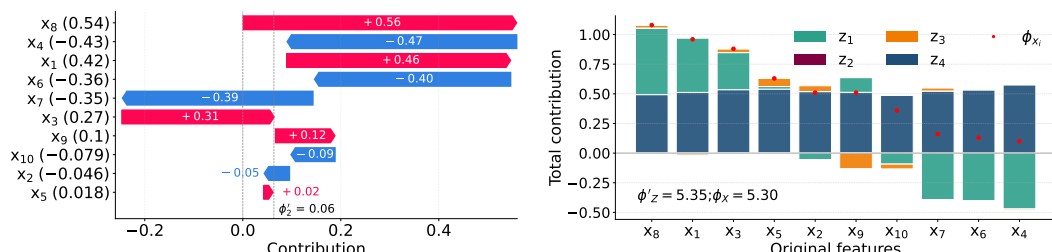

Figure 4: Illustration for OptSHAP via a simulation in which a 10-dim space is reduced to a 4-dim space: **(Left)** Local contribution for $\phi'_2$ and **(Right)** Total contribution for original features.

original features, with the values in parentheses indicating their corresponding PCA loadings. Here, features with positive loadings on $z_2$ (e.g., $x_8$ and $x_1$) receive positive contributions, while those with negative loadings (e.g., $x_6$ and $x_7$) receive negative contributions.

The right figure aggregates the allocations across all four reduced components ($z_1, z_2, z_3, z_4$). Each bar shows how the contributions from different components accumulate for a given feature, while the red dot indicates the total feature attribution $\Phi(f, \boldsymbol{x})$ obtained by OptSHAP. For example, $x_8$ receives the highest positive attributions from $z_1$, $z_3$, and $z_4$. This confirms that OptSHAP not only respects local consistency for each component but also yields intuitive overall attributions in the original feature space.

### 6.3 QUANTITATIVE EVALUATION

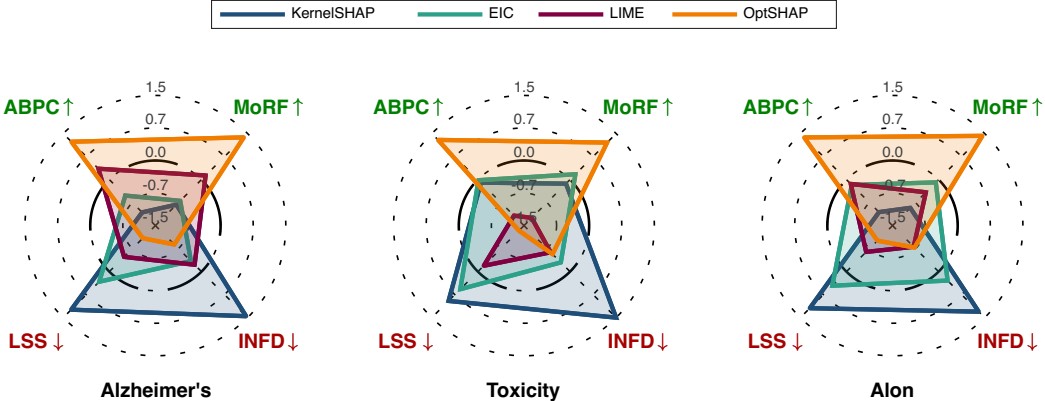

Figure 5: Comparison of OptSHAP with three baseline methods on three datasets. Metrics include MoRF and ABPC (higher is better); LSS and INFD (lower is better). Results are averaged across four tree-based models and three dimensionality reduction techniques. Detailed results for each dimensionality reduction technique and classifier are provided in Appendix E.

The quantitative results of the faithfulness evaluation are reported in Fig. 5. Our framework outperforms all baseline methods in MoRF and ABPC, demonstrating its effectiveness in identifying the most important features. This trend holds consistently across all datasets, confirming the faithfulness of our method in aligning feature attribution with model behavior. OptSHAP also achieves the lowest LSS score, showing strong stability and robustness to local perturbations. Moreover, the consistently low INFD values indicate reduced infidelity between explanations and the model's predictions, meaning that the generated attributions more faithfully capture the true decision process of the underlying model. These findings confirm that OptSHAP consistently provides more faithful (higher MoRF and ABPC), stable (lower LSS), and reliable (lower INFD) explanations compared to other attribution-based methods.

### 6.4 QUALITATIVE ANALYSIS

Fig. 6 presents a beeswarm plot showing the top five contributing features, along with the average frequency of zero-valued attributions produced by KernelSHAP, EIC, LIME, and OptSHAP. This raises a key question: *What should one expect from an effective explanation for DRbMs?*

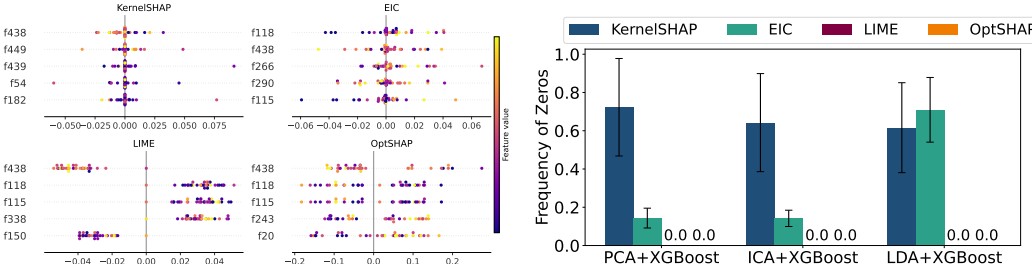

Figure 6: (**Left**) The beeswarm plot of KernelSHAP, EIC, LIME, and OptSHAP on the Alzheimer's dataset with PCA and XGBoost. (**Right**) The average frequency of zeros in the explanations, with this value being 0 for both LIME and OptSHAP.

In linear DR techniques, each component is a linear combination of all original features; consequently, each feature contributes to every principal component. Thus, we expect most original features to receive non-zero attribution scores. Zero attributions may still occur due to compensatory effects among features, but such cases are generally uncommon. Moreover, from a feature-by-feature perspective, individual features may contribute either positively or negatively to the output. Accordingly, the distribution of attribution values is expected to be roughly balanced around zero, reflecting both positive and negative contributions rather than being skewed to one side. This forms our second expectation. Importantly, this expectation is consistent with the behavior of original SHAP values (Lamane et al., 2025; Ponce-Bobadilla et al., 2024), which also exhibit such symmetric distributions, thereby confirming that our requirement is well grounded rather than ad hoc.

Fig. 6 shows that OptSHAP's explanation satisfies the above expectations. In contrast, while KernelSHAP and EIC produce symmetric explanations (meeting the second expectation), they fail to satisfy the first expectation, namely, assigning non-zero contributions to most input features. For example, KernelSHAP approximates the full 2d power set of feature combinations using a sampling scheme that assigns weights based on the subset size $s$, using the function $w(s) = \frac{d-1}{\binom{d}{s}s(d-s)}$. This weighting biases the sampling process toward smaller subsets, often resulting in sparse attributions with many zero-valued features. For a high-dimensional dataset, the large number of features $d$ can make most of the weights nearly zero, thereby causing the frequency of zero-valued attributions to be too high (over 60%). On the other hand, EIC, although having a lower zero-ratio, computing the marginal contribution based on perturbation also causes the problem that the original contribution tends to be zero. In contrast, while LIME satisfies the first expectation, it produces a skewed and narrowed contribution of each feature to one side, which fails to accurately reflect the behavior of DR techniques.

## 6.5 HYPOTHESIS TESTING FOR EFFICIENT PROPERTY

We report hypothesis testing results assessing the efficient property of OptSHAP. Specifically, we test whether the total contribution differs significantly before and after allocation. To this end, we perform a paired $t$-test, the Wilcoxon signed-rank test, and a permutation test at a 99% confidence level. The null hypothesis is $H_0 : \mu = 0$ (i.e., no statistical difference in total contribution), and the alternative hypothesis is $H_1 : \mu \neq 0$. The results in Tab. 2 indicate that OptSHAP passes all tests ($p$-value $> \alpha$). This implies there is insufficient evidence to reject $H_0$, suggesting no statistical difference in total contribution before and after allocation at the 99% confidence level.

Table 2: The $p$-value of the hypothesis testing with $\alpha = 0.01$. A $p$-value greater than $\alpha$ indicates that OptSHAP satisfies the efficient property.

| Dataset | DR Method | $t$-test | Wilcoxon | Permutation |
|---|---|---|---|---|
| Alzheimer's | PCA | 0.2075 | 0.1898 | 0.2030 |
| | ICA | 0.1628 | 0.1633 | 0.1636 |
| | LDA | 0.9485 | 0.0504 | 0.9184 |
| Toxicity | PCA | 0.2242 | 0.2500 | 0.2156 |
| | ICA | 0.4810 | 0.7344 | 0.5468 |
| | LDA | 0.7108 | 0.9922 | 0.6932 |
| Alon | PCA | 0.9875 | 0.9368 | 0.9870 |
| | ICA | 0.4036 | 0.2368 | 0.4202 |
| | LDA | 0.5579 | 0.7972 | 0.5490 |

## 6.6 THE STABILITY OF OPTSHAP

We investigate OptSHAP stability under perturbations of the model gradients by injecting Gaussian noise proportional to the mean gradient magnitude. Fig. 7 summarizes the results in terms of cosine similarity (higher is better) and normalized L2 norm (lower is better) between the original and perturbed explanations. As shown in the left panel, the cosine similarity has no change when the

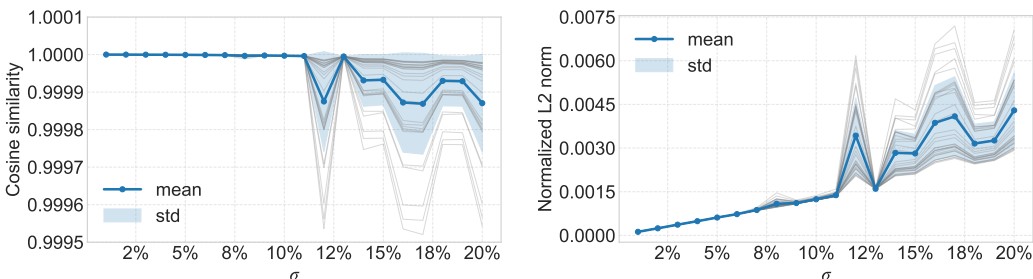

Figure 7: Stability of OptSHAP on Alzheimer's dataset under gradient-dependent Gaussian noise with strength ranging from 1% to 20% of the mean gradient magnitude.

Table 3: Stability comparison of explanation methods on Alzheimer's dataset. Reported metrics include average, median, maximum of CV, fraction of features with CV > 0.5, and median for the top 5 most important features.

| Method | mean ± std ↓ | median↓ | max↓ | fraction( 0.5)↓ | top-5 median↓ |
|---|---|---|---|---|---|
| KernelSHAP | 10.832 ± 66.034 | 2.926 | 1327.34 | 0.993 | 0.883 |
| EIC | 22.182 ± 116.599 | 4.419 | 2188.98 | 1.000 | 1.073 |
| LIME | 1.560 ± 11.652 | 0.214 | 184.32 | 0.244 | 0.043 |
| OptSHAP | **0.065 ± 0.372** | **0.011** | **5.35** | **0.020** | **0.003** |

noise is less than 10% and remains remarkably high (> 0.9996) across the entire noise range over 10%. The right panel further confirms this observation: the overall deviations of the normalized L2 error increase when noise increases, but are still small (below 0.007 at 20% noise). Together, these results show that OptSHAP produces robust explanations under gradient perturbations, ensuring reliability in practice.

Since explanation methods involve stochastic elements, repeated runs on the same explicand can yield different attributions. To quantify this variability, we generated explanations 10 times per test instance and computed the coefficient of variance (CV) across runs. Tab. 3 reports aggregated statistics of CV over the entire test set. The results highlight differences in stability. KernelSHAP and EIC show extreme variability, with mean CVs above 10, maximum values exceeding 1000 and 2000, and nearly all features above the 0.5 CV threshold. LIME is more stable, with lower mean and median CVs, though its maximum still reaches 184. By contrast, OptSHAP is highly robust: mean and median CVs are near zero, the maximum is an order of magnitude smaller, and fewer than 2% of features exceed 0.5, indicating consistent attributions for the most relevant variables.

## 7 CONCLUSIONS

In this paper, we introduce OptSHAP, which is the first optimization-based attribution method for DRbMs. This framework reprojects model-specific explanations from the reduced feature space back to the original input space through a bounded allocation transform. Accordingly, it is agnostic to the choice of DR techniques and downstream models, and scales efficiently to high-dimensional feature spaces. Evaluated on three datasets with three DR techniques and four tree-based models, OptSHAP outperforms existing methods by delivering up to 24-fold gains in stability and 2-fold improvements in fidelity, while generating explanations that align with the expected properties of interpretability for DRbMs. Moreover, our framework is compatible with nonlinear projections and deep learning models (details are provided in Appendix F), making these promising directions for future investigation. By overcoming the limitations of existing attribution methods for DRbMs and providing stable and faithful explanations, OptSHAP establishes a strong foundation for building transparent and trustworthy DR pipelines in real-world applications.

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

## THE USE OF LARGE LANGUAGE MODELS

LLMs were only used to improve the clarity and writing quality of the manuscript.

## A   ADDITIONAL THEORETICAL DETAILS

### A.1   $\tanh$ SELECTION

**Definition A.1** (Parametric Hyperbolic Tangent). Let $\alpha > 0$ and $\beta \in \mathbb{R}$. The parametric hyperbolic tangent function is defined as:

$$\tanh(\alpha x + \beta) = \frac{e^{\alpha x + \beta} - e^{-\alpha x - \beta}}{e^{\alpha x + \beta} + e^{-\alpha x - \beta}}, \quad x \in \mathbb{R}.$$

Moreover, the derivative of $\tanh(\alpha x + \beta)$ is given by:

$$\frac{d}{dx} \tanh(\alpha x + \beta) = \alpha\big(1 - \tanh^2(\alpha x + \beta)\big),$$

ensuring that $\tanh(\alpha x + \beta)$ is strictly increasing for $\alpha > 0$.

In our framework, we choose the $\tanh$ function because:

- **Capability of producing both positive and negative outputs.** Since $\tanh(\cdot)$ ranges from $-1$ to $+1$, it naturally handles contributions of opposite signs. For example in Fig. 2, if the PCA loading of cholesterol on $PC_1$ is negative while the attribution of $PC_1$ is positive, $\tanh(\cdot)$ guarantees that the propagated attribution to cholesterol is also negative. Most activation functions, such as softmax and sigmoid, do not satisfy this property.

- **Monotonicity.** The function is strictly increasing, so stronger interactions (larger gradient magnitudes $\partial_j^i$) lead to larger absolute allocations. For instance, if $\partial_{\text{height}}^{PC_1} = 0.5$ and $\partial_{\text{age}}^{PC_1} = 0.2$, then $\tanh(\alpha \cdot 0.5 + \beta)$ will always exceed $\tanh(\alpha \cdot 0.2 + \beta)$, ensuring that height receives a greater share from $PC_1$.

- **Continuity and Differentiability.** The smoothness of $\tanh(\cdot)$ ensures that our loss function (Eq. 5) is differentiable everywhere. This enables stable optimization using gradient-based methods such as Adam. Without such a smooth transform, optimization could stall or oscillate.

- **Boundedness.** The outputs of $\tanh(\cdot)$ lie strictly within $(-1, 1)$, preventing allocations from growing arbitrarily large. For example, if $\partial_j^i$ is extremely large due to scale differences across features, $\tanh(\cdot)$ saturates at $\pm 1$, thereby avoiding the situation where a single feature dominates the redistribution unfairly.

- **Symmetry around zero.** The function is odd, i.e., $\tanh(-x) = -\tanh(x)$, which aligns allocations with gradient directions. Concretely, if $\partial_j^i$ is positive, the allocation $\phi_{i \to j}^{\mathcal{T}}$ has the same sign as $\phi_i'$; if $\partial_j^i$ is negative, the allocation flips sign. This ensures that the semantics of component-feature relationships are preserved.

### A.2   SENSITIVITY ANALYSIS OF $\tanh$ FUNCTION

The parameter $\alpha$ controls the sensitivity of $\tanh(\cdot)$ to variations in the directional component $\partial_j^i$. In the limiting case, $\lim_{\alpha \to 0} \tanh\big(\alpha \cdot \partial_j^i + \beta\big) = \tanh(\beta)$, meaning that the effect of $\partial_j^i$ vanishes and the output is determined solely by the bias term $\beta$. For implementation purposes, however, directly setting $\alpha = 0$ (or arbitrarily close to zero) is not practical, as it may lead to numerical issues and hinder gradient-based optimization. Therefore, instead of enforcing $\alpha > 0$ strictly, we introduce a lower bound $\alpha_{\min} > 0$ (a small positive constant). This ensures numerical stability and preserves the desired sensitivity while keeping the code implementation simple.

Regarding the shift $\beta$ of $\tanh(.)$, we show its upper and lower bounds in the following proposition.

**Proposition A.2** (Range of $\beta$). *Given $\alpha \geq \alpha_{min}$, the shift $\beta$ of allocation transform must be in the range:*

$$-\alpha_{min} \left(\partial_j^i\right)_{min}^+ < \beta < -\alpha_{min} \left(\partial_j^i\right)_{max}^-, \tag{8}$$

*where $\left(\partial_j^i\right)_{min}^+$ and $\left(\partial_j^i\right)_{max}^-$ are the smallest positive and largest negative gradients, respectively.*

*Proof.* The parameter $\beta \in \mathbb{R}$ governs the shift of $\tanh$ and ensures the property of positive/negative preservation:

$$\text{sign}\left(\partial_j^i\right) = \text{sign}\left(\alpha\partial_j^i + \beta\right).$$

To satisfy this, $\beta$ must satisfy:

$$\alpha \cdot \partial_j^i + \beta > 0 \quad \text{if } \partial_j^i > 0, \tag{9}$$
$$\alpha \cdot \partial_j^i + \beta < 0 \quad \text{if } \partial_j^i < 0. \tag{10}$$

From 9, we deduce:

$$\beta > -\alpha \cdot \partial_j^i, \text{ for all } \partial_j^i > 0.$$

This is equivalent to:

$$\beta > \max\{-\alpha \cdot \partial_j^i | \partial_j^i > 0\} = -\min\{\alpha \cdot \partial_j^i | \partial_j^i > 0\} = -\alpha_{min}\left(\partial_j^i\right)_{min}^+,$$

where $\left(\partial_j^i\right)_{min}^+$ is the smallest positive gradient. Similarly, from 10:

$$\beta < -\alpha_{min}\left(\partial_j^i\right)_{max}^-,$$

where $\left(\partial_j^i\right)_{max}^-$ is the largest negative gradient. Combining these bounds, $\beta$ must lie in the range:

$$-\alpha_{min}\left(\partial_j^i\right)_{min}^+ < \beta < -\alpha_{min}\left(\partial_j^i\right)_{max}^-.$$

$\square$

Notably, if $\left(\partial_j^i\right)^+$ or $\left(\partial_j^i\right)^-$ is an empty set, then we set the minimum or maximum to 0, respectively. Moreover, $\beta = 0$ always satisfies these conditions.

### A.3 SIGN PRESERVATION PROPERTY

**Theorem 4.2** (Sign Preservation). *Transformation $\mathcal{T}$ satisfies the property 1 about sign preservation.*

*Proof.* We need to prove that: For all $i \in \{1, \ldots, r\}$ and $j \in \{1, \ldots, d\}$, then $\phi_i' \cdot \partial_j^i \cdot \phi_{i \rightarrow j}^{\mathcal{T}} > 0$.

By the Definition 4.1,

$$\phi_{i \rightarrow j}^{\mathcal{T}} = \tanh\left(\alpha_i \partial_j^i + \beta\right) \phi_i'.$$

Substituting this into the inequality gives:

$$\partial_j^i \cdot \tanh\left(\alpha_i \cdot \partial_j^i + \beta\right) \cdot (\phi_i')^2.$$

Note that we only consider $\phi_i' \neq 0$ because $\phi(z_i) = 0$ is a trivial case where all re-distribution will be zero. Therefore, $\phi^2(z_i) > 0$ and the sign of the product depends on:

$$\partial_j^i \cdot \tanh\left(\alpha_i \cdot \partial_j^i + \beta\right).$$

By construction, $\tanh(\cdot)$ preserves the sign of $(\alpha_i \cdot \partial_j^i + \beta)$, which aligns with $\partial_j^i$ under the conditions on $\alpha_i$ and $\beta$. Thus, the product is strictly positive, completing the proof. $\square$

## A.4 Stability under gradient-dependent Gaussian noise

**Theorem 4.4**. (Stability). *If the gradient information $J_p(\boldsymbol{x}^e)$ is perturbed by independent Gaussian noise $\delta_{ij} \sim \mathcal{N}(0, \sigma^2)$, then the expected deviation of the allocated importance $\phi_j$ satisfies:*

$$\mathbb{E}\left[\left|\phi_j - \phi_j^{noise}\right|\right] \leq \sum_{i=1}^{r} |\phi_i'| . \alpha_i . \min\left(2, \sigma\sqrt{\frac{2}{\pi}}\right).$$

*Proof.* The allocated importance $\phi_j$ and $\phi_j^{\text{noise}}$ are respectively:

$$\phi_j = \sum_{i=1}^{r} \phi_i' \tanh\left(\alpha_i \cdot \partial_j^i + \beta\right),$$

$$\phi_j^{\text{noise}} = \sum_{i=1}^{r} \phi_i' \tanh\left(\alpha_i \cdot (\partial_j^i + \delta_{ij}) + \beta\right).$$

The difference between those is:

$$\left|\phi_j - \phi_j^{\text{noise}}\right| \leq \sum_{i=1}^{r} |\phi_i'| . \left|\tanh\left(\alpha_i \cdot \partial_j^i + \beta\right) - \tanh\left(\alpha_i \cdot (\partial_j^i + \delta_{ij}) + \beta\right)\right|.$$

From $|\tanh(x) - \tanh(y)| \leq \min(2, |x - y|)$, then:

$$\left|\phi_j - \phi_j^{\text{noise}}\right| \leq \sum_{i=1}^{r} |\phi_i'| . \min(2, |\alpha_i \delta_{ij}|).$$

Because $\delta_{ij} \sim \mathcal{N}(0; \sigma^2)$, $\mathbb{E}[|\delta_{ij}|] = \sigma\sqrt{\frac{2}{\pi}}$. This implies:

$$\mathbb{E}\left[\left|\phi_j - \phi_j^{\text{noise}}\right|\right] \leq \sum_{i=1}^{r} |\phi_i'| . \alpha_i . \min\left(2, \sigma\sqrt{\frac{2}{\pi}}\right).$$

$\square$

## B Implementation Details

All experiments were conducted on RTX 3090Ti, and the code was implemented in Python 3.12.2. To ensure reproducibility, we fix the random seed to 0.

### B.1 Datasets

All of the public datasets used in our experiments were previously published, mainly focusing on the healthcare domain. These datasets are chosen due to their high dimensionality, i.e., a much larger number of features compared to the number of samples, which aligns with the objectives of this work. Moreover, they are split into training and testing sets with a ratio of 8:2. The details of the datasets are listed in the following:

- **Alzheimer's** (Fontanella, 2022), 451 features and 171 samples. Also referred to as the DAR-WIN dataset, this collection contains handwriting samples from 174 individuals, with the goal of classifying Alzheimer's disease patients versus healthy controls. It was designed to facilitate the development of machine learning techniques for Alzheimer's disease prediction based on handwriting analysis.
- **Toxicity** (Gül & RAHIM, 2021), 1203 features and 174 samples. This dataset contains 171 molecules engineered to target functional domains of the core circadian clock protein CRY1, which regulates circadian rhythms. Among them, 56 molecules are labeled as toxic, while the remaining are non-toxic.
- **Alon** (Alon et al., 1999), 2000 features and 62 samples. The Alon colon cancer dataset, introduced by *Alon et al.*, comprises expression levels of 2000 genes across 62 samples. 40 tumor biopsies and 22 normal colon tissue biopsies were obtained from the same patients.

## B.2 DR Techniques

We use three popular DR techniques, namely PCA, ICA, and LDA. All are available in `sklearn` library. Detailed implementation for these techniques is presented as follows:

- **Principal Component Analysis (PCA) (Abdi & Williams, 2010)**: The number of components $r$ is selected as the smallest integer such that the cumulative variance exceeds a predefined threshold (e.g., 95%). Then, the reduced representation $\mathcal{Z} \in \mathbb{R}^{n \times r}$ is obtained via $\mathcal{Z} = \mathcal{X}W_r$, where $W_r \in \mathbb{R}^{d \times r}$ is the loading matrix composed of the top $r$ eigenvectors.

- **Independent Component Analysis (ICA) (Hyvärinen et al., 2009)**: This is deployed by using the FastICA algorithm to extract statistically independent components from the input data. We set the number of components as $r = 0.05d$, where $d$ is the number of original features. FastICA is configured with unit-variance whitening, a maximum of 1000 iterations, and a fixed random seed for reproducibility (i.e, 42). The data is then projected to an $r$-dimensional space, yielding the reduced representation $\mathcal{Z} \in \mathbb{R}^{n \times r}$. The corresponding loading matrix $W_r \in \mathbb{R}^{d \times r}$ is derived from the learned independent components.

- **Linear Discriminant Analysis (LDA) (Balakrishnama & Ganapathiraju, 1998)**: We implement LDA to reduce supervised dimensionality by maximizing class separability. Given a dataset $\mathcal{X}$ and corresponding class labels $\mathbf{y}$, we compute the number of components $r = \min(C-1, d)$, where $C$ is the number of distinct classes and $d$ is the number of original features. LDA is then fitted to the labeled data $(\mathcal{X}, \mathbf{y})$, and the input is projected into an $r$-dimensional space, resulting in a reduced representation $\mathcal{Z} \in \mathbb{R}^{n \times r}$. The transformation matrix $W_r \in \mathbb{R}^{d \times r}$ is constructed from the top $r$ linear discriminants (i.e., eigenvectors of the class scatter matrix), accessible via the learned scaling matrix.

## B.3 Classifiers

Regarding learning models, we adopt four widely used ensemble methods tailored for binary classification tasks: XGBoost, LightGBM, CatBoost, and Random Forest.

- **XGBoost** (Chen & Guestrin, 2016) is instantiated via *XGBClassifier*, with the objective function set to *binary logistic loss* and the evaluation metric to *logloss*.

- **CatBoost** (Prokhorenkova et al., 2018) is implemented using *CatBoostClassifier*, configured with the *logloss* (binary cross-entropy) loss function and verbose output disabled.

- **LightGBM** (Ke et al., 2017) leverages *LGBMClassifier*, with the objective function set to *binary* and the evaluation metric to *binary logloss*.

- **Random Forest** (Breiman, 2001) uses *RandomForestClassifier* with 100 estimators, suitable for binary classification tasks.

Each model is trained on the reduced feature space produced from one of the aforementioned DR techniques.

## B.4 Feature attribution explanation methods

We implement and compare our framework, OptSHAP, with three other feature attribution methods: KernelSHAP, LIME, and EIC. All methods were applied to the same trained model and test set, using consistent settings to ensure fair comparison. For each explicand (test sample), the explanation is generated for the predicted class.

- **KernelSHAP** (Lundberg & Lee, 2017): The model-agnostic KernelSHAP method is implemented using the implementation provided in the `shap` library. The explainer is initialized using the training set $\mathbf{X}_{\text{train}}$ as the background distribution (without sampling), and explanations are computed on $\mathbf{X}_{\text{test}} \subset \mathcal{X}$. For each test sample, SHAP values are generated using 1000 sampled coalitions.

- **LIME** (Ribeiro et al., 2016): The LIME explainer is implemented using the `lime` library. It is initialized using the full training data $\mathbf{X}_{\text{train}}$ as the background distribution, while feature names and class labels are provided. For each test sample $\boldsymbol{x} \in \mathbf{X}_{\text{test}}$, LIME generates an

explanation by fitting a local interpretable linear model on 1000 perturbed samples around $x$, using the DRBM. We configure the number of features in each explanation to match the input dimensionality (i.e., `num_features = d`). The local surrogate model used for attribution is the default *Ridge regression*.

- **EIC** (Strumbelj & Kononenko, 2010): The EIC method estimates local feature attributions based on perturbation-based marginal effects. Specifically, for each sample $x \in \mathbf{X}_{\text{test}}$, EIC estimates the contribution of feature $i$ as the *expected decrease in model prediction* when feature $i$ is perturbed, while conditioning on a random subset of the remaining features. To do so, we run Monte Carlo randomized perturbation trials for each test sample $x$. In each trial, a random binary mask $\mathbf{s} \in \{0,1\}^d$ selects a subset of features to keep from $x$, while the rest are filled from a randomly sampled background point $x' \in \mathbf{X}_{\text{train}}$. For each feature $i \in \text{supp}(\mathbf{s})$, we measure the influence as the difference in the model output when $x_i$ is replaced with a background value $x_i'$, formally: $\text{contrib}_i = \mathbb{E}_{\mathbf{s},x'}[f(\tilde{x}) - f(\tilde{x}_{-i})]$, where $\tilde{x}$ is the mixed input and $\tilde{x}_{-i}$ replaces feature $i$ with $x_i'$. The final explanation is the average over $k$ trials.

- **OptSHAP**: We implement Opt-SHAP as described in Alg. 1 and apply it to the entire projected test set $\mathbf{Z}_{\text{test}}$, without subsampling. The reduced-space SHAP values $\phi'$ are computed using the `shap.TreeExplainer`, while the Jacobian matrix $\mathbf{J}$ is obtained from the DR transformation (e.g., PCA, ICA, LDA) fitted on the training data $\mathbf{X}_{\text{train}}$. The optimization is performed using the *Adam* optimizer, with learning rate set to 0.02 and up to 300 iterations. Parameters $(\alpha_1, \ldots, \alpha_r, \beta)$ are optimized in constrained space: $\alpha_i \geq 10^{-3}$, while $\beta$ is restricted by data-dependent lower/upper limits. All $\alpha_i$ are initialized to the minimum, and $\beta$ is initialized at zero.

## C  EVALUATION METRICS

**Definition C.1.** Let $g : \mathbb{R}^d \to \mathbb{R}$ be a black-box model, which predicts the output $g(x)$ for an input $x \in \mathbb{R}^d$. A feature attribution explanation is a function $\Phi : \mathcal{G} \times \mathbb{R}^d \to \mathbb{R}^d$, that given predictor $g$ and a test point $x$, assigns importance scores $\Phi(g, x)$ to the input features, known as an *explanation*.

**Faithfulness evaluation.** As mentioned in Sec. 5, we evaluate the faithfulness of explanation methods via MoRF, ABPC, LSS, and INFD. Now, given a black-box function $g$, an explanation functional $\Phi$, a random variable $I$ with probability measure $\mu_I$, which represents meaningful perturbations of interest. The formulation of MoRF, LeRF, ABPC, and INFD metrics is presented as follows:

$$\text{MoRF}(x) = \frac{1}{d+1} \sum_{k=0}^{d} \left( g(x_{\text{MoRF}}^{(0)}) - g(x_{\text{MoRF}}^{(k)}) \right), \tag{11}$$

$$\text{LeRF}(x) = \frac{1}{d+1} \sum_{k=0}^{d} \left( g(x_{\text{LeRF}}^{(0)}) - g(x_{\text{LeRF}}^{(k)}) \right), \tag{12}$$

$$\text{ABPC}(x) = \frac{1}{d+1} \sum_{k=0}^{d} \left( g(x_{\text{LeRF}}^{(k)}) - g(x_{\text{MoRF}}^{(k)}) \right), \tag{13}$$

$$\text{INFD}(x) = \mathbb{E}_{I \sim \mu_I} \left( I^T \Phi(g, x) - (g(x) - g(x - I))^2 \right). \tag{14}$$

Here, $d$ is the number of features, $x^{(0)}$ is the original input, $x_{\text{MoRF}}^{(k)}$ and $x_{\text{LeRF}}^{(k)}$ denotes a perturbed version of the input in which the top-$k$ most and bottom-$k$ least relevant features are masked, respectively. In the INFD evaluation (Yeh et al., 2019), we use the method of difference from a noisy baseline $I = x - w_o$, where $w_o = x_o + \epsilon$, for some zero-mean random vector $\epsilon \sim \mathcal{N}(0, \sigma^2)$.

**Stability metrics.** In addition to the faithfulness metrics, we assess the *stability* of explanations under stochastic perturbations, repeated runs, or noisy gradients. Given two explanations $\Phi(g, x)$ and $\Phi'(g, x)$ for the same explicand $x$, we define the following:

$$\text{CosineSim}(\Phi, \Phi') = \frac{\langle \Phi(g, \boldsymbol{x}), \ \Phi'(g, \boldsymbol{x}) \rangle}{\|\Phi(g, \boldsymbol{x})\|_2 \cdot \|\Phi'(g, \boldsymbol{x})\|_2}, \tag{15}$$

$$\text{NormL2}(\Phi, \Phi') = \frac{\|\Phi(g, \boldsymbol{x}) - \Phi'(g, \boldsymbol{x})\|_2}{\|\Phi(g, \boldsymbol{x})\|_2 + \Phi'(g, \boldsymbol{x})\|_2}, \tag{16}$$

where $\langle \cdot, \cdot \rangle$ denotes the inner product and $\| \cdot \|_2$ the Euclidean norm. Cosine similarity (Eq. 15) measures directional consistency between two explanations, with values close to 1 indicating strong alignment. The normalized L2 norm (Eq. 16) measures the relative deviation in magnitude between explanations, with smaller values indicating more robust attributions.

To quantify variability across multiple stochastic runs, we adopt the *coefficient of variance* (CV), defined for each feature $j$ as

$$\text{CV}_j = \frac{\sigma\left(\{\phi_j^{(1)}, \phi_j^{(2)}, \ldots, \phi_j^{(R)}\}\right)}{\left|\mu\left(\{\phi_j^{(1)}, \phi_j^{(2)}, \ldots, \phi_j^{(R)}\}\right)\right|}, \tag{17}$$

where $\phi_j^{(r)}$ denotes the attribution score of feature $j$ in the $r$-th run, and $\mu(\cdot)$ and $\sigma(\cdot)$ denote the mean and standard deviation across $R$ repeated runs. A higher CV indicates stronger instability, while a CV close to zero indicates highly reproducible attributions.

## D  G-DEEPSHAP IS NOT SUITABLE FOR DRBMS

As discussed in Sec. 2, we show that G-DeepSHAP is not suitable in the context of DRbMs. Formally, G-DeepSHAP computes the attribution for an explicand $\mathbf{x}^e$ with respect to a baseline set $D$ as:

$$\phi(f \circ p, \boldsymbol{x}^e) = \frac{1}{|D|} \sum_{\boldsymbol{x}^b \in D} \phi(f \circ p, \boldsymbol{x}^e, \boldsymbol{x}^b) = \frac{1}{|D|} \sum_{\boldsymbol{x}^b \in D} \widehat{\phi}(p, \boldsymbol{x}^e, \boldsymbol{x}^b) \frac{\widehat{\phi}(f, \boldsymbol{z}^e, \boldsymbol{z}^b)}{p(\boldsymbol{x}^e) - p(\boldsymbol{x}^b)} \tag{18}$$

$$= \frac{1}{|D|} \sum_{\boldsymbol{x}^b \in D} \mathbf{diag}(x^e - x^b) W \cdot \frac{\widehat{\phi}(f, \boldsymbol{z}^e, \boldsymbol{z}^b)}{(z^e - z^b)} \tag{19}$$

Here, $\widehat{\phi}(f, \boldsymbol{z}^e, \boldsymbol{z}^b)$ is the explanation in the reduced space. This expression makes clear why G-DeepSHAP is problematic in the DRbM setting. The term $(z^e - z^b)$ appears in the denominator, and nothing prevents it from approaching zero on any coordinate. In practice, this means that whenever two latent encodings $\mathbf{z}^e$ and $\mathbf{z}^b$ are close, the attribution can blow up to arbitrarily large magnitudes. Since DRbMs project high-dimensional inputs $X$ into a lower-dimensional latent space $Z$ before classification, different inputs may be projected to nearby points in $Z$. As a consequence, cases with $\mathbf{z}^e \approx \mathbf{z}^b$ naturally arise with high frequency. Moreover, because G-DeepSHAP averages over a set of baselines $D$, the probability of encountering such near-zero differences only increases, thereby amplifying the instability.

This theoretical issue is clearly reflected in empirical results. Figure 8 shows both a boxplot and a heatmap of G-DeepSHAP attributions computed on DRbM data. The distributions are highly skewed, with certain features displaying extreme values far exceeding reasonable scales. In the heatmap, entire rows are dominated by intense colors corresponding to very large positive or negative attributions, even though the explicand and baselines differ only slightly in the latent space. Such "blow-ups" undermine the interpretability of the method: a small perturbation in baseline selection disproportionately dictates the final explanation, masking the genuine structure of the model.

Taken together, both the analytical form and the empirical evidence indicate that G-DeepSHAP is not suitable for DRbMs. The instability of its attributions not only reduces their reliability but also risks misleading users about the true behavior of the underlying model. Prior work has emphasized that unstable or manipulable explanations can severely compromise trust in post-hoc methods (Slack et al., 2020), and stability has been recognized as a key desideratum for meaningful feature attributions (Xue et al., 2023). Moreover, recent critiques of Shapley-based explanations highlight that when the attribution values deviate drastically from reasonable scales, the interpretations cease to reflect the underlying model behavior (Huang & Marques-Silva, 2024). These observations suggest that in the space of DRbMs, where attribution "blow-ups" are frequent, G-DeepSHAP is not a suitable explanation method.

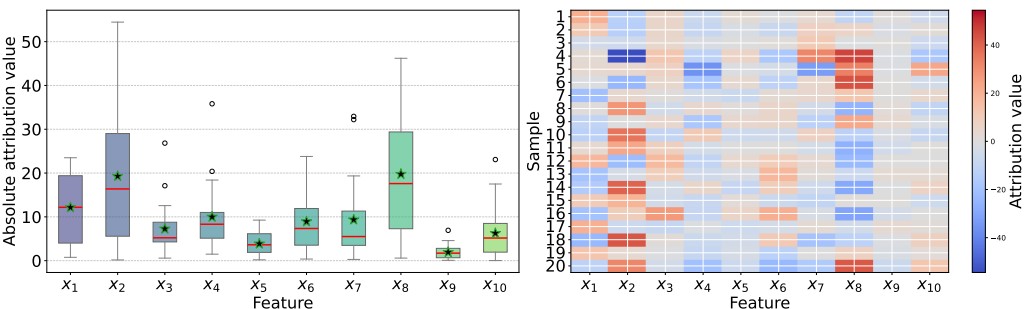

Figure 8: Boxplot and heatmap of G-DeepSHAP attributions on DRbM data, illustrating unstable and exploding values.

# E ADDITIONAL EXPERIMENT RESULTS

Table 4: Results for the Alzheimer's dataset averaged on 3 dimensionality reduction techniques (Best result in each row is in bold).

| Model | Method | MoRF | ABPC | LSS | INFD |
|---|---|---|---|---|---|
| XGBoost | KernelSHAP | $0.0180 \pm 0.0183$ | $-0.0093 \pm 0.0277$ | $0.0044 \pm 0.0000$ | $2.0304 \pm 0.9019$ |
| | EIC | $0.0211 \pm 0.0389$ | $0.0025 \pm 0.0406$ | $0.0044 \pm 0.0000$ | $0.7520 \pm 1.2704$ |
| | LIME | $0.0366 \pm 0.0279$ | $0.0271 \pm 0.0262$ | $0.0027 \pm 0.0002$ | $0.8288 \pm 0.3901$ |
| | OptSHAP | $\mathbf{0.0595 \pm 0.0431}$ | $\mathbf{0.0566 \pm 0.0363}$ | $\mathbf{0.0022 \pm 0.0005}$ | $\mathbf{0.0553 \pm 0.0238}$ |
| LightGBM | KernelSHAP | $0.0221 \pm 0.0302$ | $-0.0019 \pm 0.0330$ | $0.0044 \pm 0.0000$ | $2.2965 \pm 1.5589$ |
| | EIC | $0.0209 \pm 0.0363$ | $0.0159 \pm 0.0280$ | $0.0037 \pm 0.0007$ | $1.0361 \pm 1.3444$ |
| | LIME | $0.0323 \pm 0.0357$ | $0.0246 \pm 0.0229$ | $0.0028 \pm 0.0002$ | $0.7049 \pm 0.4822$ |
| | OptSHAP | $\mathbf{0.0548 \pm 0.0359}$ | $\mathbf{0.0489 \pm 0.0270}$ | $\mathbf{0.0023 \pm 0.0005}$ | $\mathbf{0.0520 \pm 0.0318}$ |
| CatBoost | KernelSHAP | $-0.0022 \pm 0.0175$ | $-0.0241 \pm 0.0246$ | $0.0044 \pm 0.0000$ | $2.5874 \pm 0.9250$ |
| | EIC | $0.0126 \pm 0.0357$ | $0.0059 \pm 0.0402$ | $0.0034 \pm 0.0003$ | $0.1054 \pm 0.0392$ |
| | LIME | $0.0137 \pm 0.0220$ | $0.0133 \pm 0.0162$ | $0.0025 \pm 0.0003$ | $0.6934 \pm 0.4813$ |
| | OptSHAP | $\mathbf{0.0296 \pm 0.0236}$ | $\mathbf{0.0289 \pm 0.0174}$ | $\mathbf{0.0023 \pm 0.0005}$ | $\mathbf{0.0514 \pm 0.0256}$ |
| Random Forest | KernelSHAP | $-0.0254 \pm 0.0597$ | $-0.0184 \pm 0.0254$ | $0.0044 \pm 0.0000$ | $3.3883 \pm 0.6985$ |
| | EIC | $-0.0349 \pm 0.0781$ | $-0.0288 \pm 0.0295$ | $0.0034 \pm 0.0002$ | $0.6410 \pm 1.0460$ |
| | LIME | $-0.0148 \pm 0.0707$ | $0.0094 \pm 0.0295$ | $0.0027 \pm 0.0002$ | $0.8234 \pm 0.3902$ |
| | OptSHAP | $\mathbf{-0.0043 \pm 0.0683}$ | $\mathbf{0.0191 \pm 0.0285}$ | $\mathbf{0.0018 \pm 0.0013}$ | $\mathbf{0.0533 \pm 0.0265}$ |

Table 5: Results for the Alzheimer's dataset averaged on 4 classifiers (Best result in each row is in bold).

| DR Technique | Method | MoRF | ABPC | LSS | INFD |
|---|---|---|---|---|---|
| PCA | KernelSHAP | $-0.0344 \pm 0.0489$ | $-0.0369 \pm 0.0152$ | $0.0044 \pm 0.0000$ | $2.0080 \pm 1.0011$ |
| | EIC | $-0.0545 \pm 0.0561$ | $-0.0492 \pm 0.0320$ | $0.0034 \pm 0.0002$ | $0.0811 \pm 0.0414$ |
| | LIME | $-0.0316 \pm 0.0519$ | $-0.0088 \pm 0.0167$ | $0.0026 \pm 0.0003$ | $0.4547 \pm 0.4359$ |
| | OptSHAP | $\mathbf{-0.0154 \pm 0.0527}$ | $\mathbf{0.0146 \pm 0.0230}$ | $\mathbf{0.0026 \pm 0.0002}$ | $\mathbf{0.0651 \pm 0.0206}$ |
| ICA | KernelSHAP | $0.0183 \pm 0.0129$ | $-0.0251 \pm 0.0147$ | $0.0044 \pm 0.0000$ | $1.9250 \pm 0.7922$ |
| | EIC | $0.0321 \pm 0.0223$ | $0.0096 \pm 0.0249$ | $0.0034 \pm 0.0002$ | $0.0970 \pm 0.0375$ |
| | LIME | $0.0363 \pm 0.0164$ | $0.0222 \pm 0.0082$ | $0.0028 \pm 0.0001$ | $0.7260 \pm 0.4243$ |
| | OptSHAP | $\mathbf{0.0467 \pm 0.0153}$ | $\mathbf{0.0313 \pm 0.0104}$ | $\mathbf{0.0026 \pm 0.0001}$ | $\mathbf{0.0547 \pm 0.0288}$ |
| LDA | KernelSHAP | $0.0256 \pm 0.0144$ | $0.0217 \pm 0.0134$ | $0.0044 \pm 0.0000$ | $3.7939 \pm 0.5978$ |
| | EIC | $0.0372 \pm 0.0160$ | $0.0361 \pm 0.0160$ | $0.0038 \pm 0.0002$ | $1.7227 \pm 1.3952$ |
| | LIME | $0.0461 \pm 0.0186$ | $0.0425 \pm 0.0155$ | $0.0026 \pm 0.0003$ | $1.1071 \pm 0.0611$ |
| | OptSHAP | $\mathbf{0.0735 \pm 0.0321}$ | $\mathbf{0.0693 \pm 0.0290}$ | $\mathbf{0.0012 \pm 0.0008}$ | $\mathbf{0.0271 \pm 0.0191}$ |

Table 6: Results for the Toxicity dataset averaged on 3 dimensionality reduction techniques (Best result in each row is in bold).

| Model | Method | MoRF | ABPC | LSS | INFD |
|---|---|---|---|---|---|
| XGBoost | KernelSHAP | -0.0545 ± 0.0544 | -0.0392 ± 0.0358 | 0.0016 ± 0.0000 | 0.8352 ± 0.9621 |
| | EIC | -0.0284 ± 0.0311 | -0.0124 ± 0.0324 | 0.0014 ± 0.0001 | 0.7331 ± 0.8543 |
| | LIME | -0.0344 ± 0.0367 | -0.0018 ± 0.0171 | 0.0012 ± 0.0000 | **0.0477 ± 0.0376** |
| | OptSHAP | **-0.0162 ± 0.0434** | **0.0157 ± 0.0130** | **0.0010 ± 0.0002** | 0.0691 ± 0.0521 |
| LightGBM | KernelSHAP | -0.0164 ± 0.0349 | -0.0193 ± 0.0219 | 0.0016 ± 0.0000 | 1.5242 ± 1.3618 |
| | EIC | 0.0069 ± 0.0390 | 0.0210 ± 0.0343 | 0.0014 ± 0.0001 | 0.8762 ± 1.1327 |
| | LIME | -0.0119 ± 0.0288 | -0.0123 ± 0.0227 | 0.0011 ± 0.0000 | **0.0200 ± 0.0099** |
| | OptSHAP | **0.0280 ± 0.0602** | **0.0293 ± 0.0412** | **0.0011 ± 0.0001** | 0.0493 ± 0.0307 |
| CatBoost | KernelSHAP | -0.0174 ± 0.0338 | -0.0159 ± 0.0123 | 0.0016 ± 0.0000 | 1.5803 ± 1.8537 |
| | EIC | -0.0033 ± 0.0510 | 0.0067 ± 0.0403 | 0.0013 ± 0.0001 | 0.5161 ± 0.6541 |
| | LIME | -0.0164 ± 0.0378 | -0.0146 ± 0.0184 | 0.0011 ± 0.0000 | **0.0044 ± 0.0019** |
| | OptSHAP | **0.0148 ± 0.0503** | **0.0171 ± 0.0357** | **0.0010 ± 0.0002** | 0.0197 ± 0.0117 |
| Random Forest | KernelSHAP | -0.0561 ± 0.0528 | -0.0113 ± 0.0116 | 0.0017 ± 0.0000 | 2.6526 ± 2.4976 |
| | EIC | -0.0712 ± 0.0803 | -0.0467 ± 0.0568 | 0.0013 ± 0.0000 | 1.3536 ± 1.8449 |
| | LIME | -0.0519 ± 0.0568 | -0.0011 ± 0.0162 | 0.0011 ± 0.0000 | **0.0153 ± 0.0106** |
| | OptSHAP | **-0.0343 ± 0.0696** | **0.0007 ± 0.0341** | **0.0008 ± 0.0005** | 0.0307 ± 0.0227 |

Table 7: Results for the Toxicity dataset averaged on 4 classifiers (Best result in each row is in bold).

| DR Technique | Method | MoRF | ABPC | LSS | INFD |
|---|---|---|---|---|---|
| PCA | KernelSHAP | -0.0413 ± 0.0245 | -0.0097 ± 0.0097 | 0.0017 ± 0.0000 | 0.6533 ± 0.4814 |
| | EIC | -0.0539 ± 0.0421 | -0.0417 ± 0.0288 | 0.0013 ± 0.0001 | 0.0479 ± 0.0173 |
| | LIME | -0.0459 ± 0.0326 | -0.0147 ± 0.0106 | 0.0011 ± 0.0000 | **0.0209 ± 0.0214** |
| | OptSHAP | **-0.0368 ± 0.0289** | **-0.0139 ± 0.0155** | **0.0012 ± 0.0000** | 0.0410 ± 0.0186 |
| ICA | KernelSHAP | -0.0856 ± 0.0321 | -0.0492 ± 0.0243 | 0.0016 ± 0.0000 | 0.2987 ± 0.1828 |
| | EIC | -0.0608 ± 0.0508 | -0.0238 ± 0.0466 | 0.0013 ± 0.0000 | 0.1063 ± 0.0519 |
| | LIME | -0.0643 ± 0.0206 | -0.0157 ± 0.0254 | 0.0011 ± 0.0000 | **0.0394 ± 0.0303** |
| | OptSHAP | **-0.0451 ± 0.0289** | **0.0032 ± 0.0083** | **0.0012 ± 0.0000** | 0.0779 ± 0.0381 |
| LDA | KernelSHAP | 0.0186 ± 0.0094 | -0.0054 ± 0.0049 | 0.0016 ± 0.0000 | 3.9922 ± 1.4275 |
| | EIC | 0.0427 ± 0.0183 | 0.0419 ± 0.0191 | 0.0015 ± 0.0000 | 2.4551 ± 0.9444 |
| | LIME | 0.0242 ± 0.0046 | 0.0080 ± 0.0069 | 0.0011 ± 0.0000 | **0.0051 ± 0.0043** |
| | OptSHAP | **0.0762 ± 0.0252** | **0.0578 ± 0.0205** | **0.0006 ± 0.0004** | 0.0076 ± 0.0048 |

Table 8: Results for the Alon dataset averaged on 3 dimensionality reduction techniques (Best result in each row is in bold).

| Model | Method | MoRF | ABPC | LSS | INFD |
|---|---|---|---|---|---|
| XGBoost | KernelSHAP | 0.0166 ± 0.0595 | -0.0032 ± 0.0259 | 0.0010 ± 0.0000 | 0.4633 ± 0.4046 |
| | EIC | 0.0176 ± 0.0590 | -0.0052 ± 0.0249 | 0.0009 ± 0.0001 | 0.1086 ± 0.0759 |
| | LIME | 0.0121 ± 0.0481 | -0.0106 ± 0.0168 | 0.0007 ± 0.0000 | **0.0091 ± 0.0045** |
| | OptSHAP | **0.0349 ± 0.0648** | **0.0241 ± 0.0299** | **0.0005 ± 0.0001** | 0.0213 ± 0.0202 |
| LightGBM | KernelSHAP | 0.0433 ± 0.0509 | 0.0109 ± 0.0198 | 0.0010 ± 0.0000 | 0.5626 ± 0.3884 |
| | EIC | **0.0506 ± 0.0567** | **0.0238 ± 0.0374** | 0.0009 ± 0.0001 | 0.1098 ± 0.0676 |
| | LIME | 0.0392 ± 0.0552 | 0.0063 ± 0.0329 | 0.0007 ± 0.0000 | **0.0242 ± 0.0198** |
| | OptSHAP | 0.0418 ± 0.0863 | 0.0148 ± 0.0682 | **0.0005 ± 0.0001** | 0.0346 ± 0.0241 |
| CatBoost | KernelSHAP | -0.0175 ± 0.0151 | -0.0056 ± 0.0058 | 0.0010 ± 0.0000 | 0.4665 ± 0.4414 |
| | EIC | -0.0176 ± 0.0280 | -0.0119 ± 0.0138 | 0.0009 ± 0.0001 | 0.0605 ± 0.0475 |
| | LIME | -0.0148 ± 0.0066 | -0.0043 ± 0.0114 | 0.0007 ± 0.0000 | **0.0160 ± 0.0115** |
| | OptSHAP | **0.0019 ± 0.0144** | **0.0156 ± 0.0289** | **0.0005 ± 0.0001** | 0.0336 ± 0.0372 |
| Random Forest | KernelSHAP | **0.0537 ± 0.0605** | **0.0346 ± 0.0393** | 0.0010 ± 0.0000 | 0.4252 ± 0.3752 |
| | EIC | 0.0499 ± 0.0603 | 0.0317 ± 0.0448 | 0.0009 ± 0.0001 | 0.0771 ± 0.0391 |
| | LIME | 0.0435 ± 0.0527 | 0.0237 ± 0.0456 | 0.0007 ± 0.0000 | **0.0203 ± 0.0126** |
| | OptSHAP | 0.0495 ± 0.0382 | 0.0333 ± 0.0265 | **0.0002 ± 0.0003** | 0.0332 ± 0.0205 |

Table 9: Results for the Alon dataset averaged on 4 classifiers (Best result in each row is in bold).

| DR Technique | Method | MoRF | ABPC | LSS | INFD |
|---|---|---|---|---|---|
| PCA | KernelSHAP | $0.0407 \pm 0.0633$ | $0.0158 \pm 0.0182$ | $0.0010 \pm 0.0000$ | $0.2153 \pm 0.1025$ |
| | EIC | $0.0400 \pm 0.0732$ | $0.0178 \pm 0.0393$ | $0.0009 \pm 0.0001$ | $0.0582 \pm 0.0365$ |
| | LIME | $0.0411 \pm 0.0556$ | $0.0199 \pm 0.0184$ | $0.0007 \pm 0.0000$ | $\mathbf{0.0262 \pm 0.0156}$ |
| | OptSHAP | $\mathbf{0.0799 \pm 0.0633}$ | $\mathbf{0.0656 \pm 0.0256}$ | $\mathbf{0.0006 \pm 0.0000}$ | $0.0571 \pm 0.0173$ |
| ICA | KernelSHAP | $\mathbf{0.0163 \pm 0.0725}$ | $0.0108 \pm 0.0478$ | $0.0010 \pm 0.0000$ | $1.0432 \pm 0.0615$ |
| | EIC | $0.0159 \pm 0.0707$ | $\mathbf{0.0112 \pm 0.0497}$ | $0.0009 \pm 0.0000$ | $0.0659 \pm 0.0383$ |
| | LIME | $0.0141 \pm 0.0624$ | $0.0099 \pm 0.0440$ | $0.0007 \pm 0.0000$ | $\mathbf{0.0140 \pm 0.0142}$ |
| | OptSHAP | $0.0157 \pm 0.0474$ | $0.0102 \pm 0.0092$ | $\mathbf{0.0002 \pm 0.0001}$ | $0.0166 \pm 0.0217$ |
| LDA | KernelSHAP | $0.0150 \pm 0.0105$ | $\mathbf{0.0010 \pm 0.0044}$ | $0.0010 \pm 0.0000$ | $0.1797 \pm 0.0605$ |
| | EIC | $\mathbf{0.0195 \pm 0.0035}$ | $-0.0002 \pm 0.0035$ | $0.0009 \pm 0.0000$ | $0.1429 \pm 0.0689$ |
| | LIME | $0.0049 \pm 0.0094$ | $-0.0184 \pm 0.0113$ | $0.0007 \pm 0.0000$ | $\mathbf{0.0119 \pm 0.0075}$ |
| | OptSHAP | $0.0141 \pm 0.0212$ | $0.0005 \pm 0.0278$ | $\mathbf{0.0004 \pm 0.0002}$ | $0.0183 \pm 0.0190$ |

# F    EXTEND OptSHAP TO NONLINEAR DR TECHNIQUES

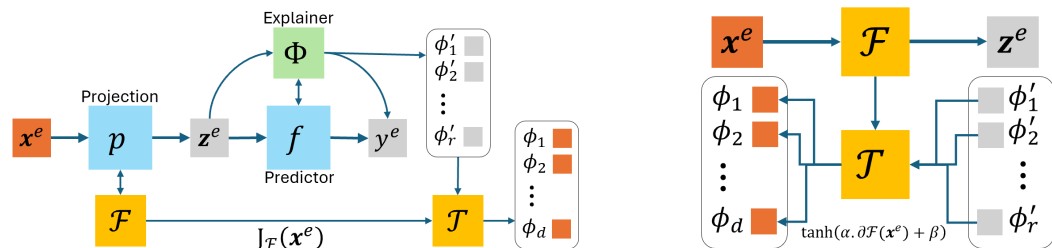

Figure 9: General (**Left**) and Internal (**Right**) view of the OptSHAP framework in case the projection can be nonlinear.

Interpreting the mapping between the original data $\mathcal{X}$ and the reduced data $\mathcal{Z}$ is a critical issue. For such a purpose, in Fig. 9, a new block $\mathcal{F}$ is proposed for approximating the mapping between $\mathcal{X}$ and $\mathcal{Z}$. This approximation is a basis for explaining the contribution of high-dimensional features $\mathbf{x}_1, \mathbf{x}_2, \ldots, \mathbf{x}_d$ towards reduced representations $\mathbf{z}_1, \mathbf{z}_2, \ldots, \mathbf{z}_r$. To make $\mathcal{F}$ strong enough to cover a diversity of DR techniques, a general form is proposed in such a way that both linear and non-linear techniques can fall under its general form. We express $\mathcal{F}_i$ as follow:

$$\mathcal{F}_i(\mathbf{x}_1, \mathbf{x}_2, \ldots, \mathbf{x}_d) = g_i\left(\sum_{j=1}^{d} h_{ij}(\mathbf{x}_j)\right), i = \overline{1, r}.$$

In this expression, $h_{ij}(\mathbf{x}_j)$ transforms individual source feature $\mathbf{x}_j$ and $g_i(\cdot)$ sums them together, possibly with new non-linearities added. With such a general form, $\mathcal{F}$ can mimic a range of DR techniques with proper selection of $h_{ij}$ and $g_i$. Specifically, there are two scenarios considered:

In the case of linear approaches, $\mathcal{F}_i$ reduces to a simple linear combination, i.e., $h_{ij}(\mathbf{x}_j) = w_{ji}\mathbf{x}_j$ and $g_i$ an identity function, and thus: $\mathcal{F}_i(\mathbf{x}_1, \mathbf{x}_2, \ldots, \mathbf{x}_d) = \sum_{j=1}^{d} w_{ji}\mathbf{x}_j$. This corresponds to methods such as PCA, ICA, or LDA. The simplicity of such a form encourages interpretability at a global level, in that each feature in reduced form, $\mathbf{z}_i$ is a weighted combination of the original features.

For such techniques that utilize non-linear relations in an attempt to capture complex structures in data, $\mathcal{F}_i$ will have to extend beyond linear mixtures. In such cases, $h_{ij}(\mathbf{x}_j)$ can denote nonlinear transformations, and $g_i(\cdot)$ can introduce nonlinearities in addition to any present in $h$. For instance, in a Single-Layer Perceptron (SLP), $h_{ij}(\mathbf{x}_j)$ is $w_{ji}\mathbf{x}_j + b_i$ and $g_i(\cdot)$ is a non-linear activation function such as ReLU or Sigmoid. The resulting formulation is:

$$\mathcal{F}_i(\mathbf{x}_1, \mathbf{x}_2, \ldots, \mathbf{x}_d) = \sigma\left(\sum_{j=1}^{d} w_{ji}\mathbf{x}_j + b_i\right).$$

The flexibility of $\mathcal{F}$ comes from the fact that most of the DR methods can be approximated by defining $h_{ij}(\mathbf{x}_j)$ and $g_i(\cdot)$ appropriately. As an example, kernel PCA can be incorporated by defining $h_{ij}(\mathbf{x}_j)$ as a kernel function $\kappa(\mathbf{x}_j, \mathbf{x}_k)$ which maps the data into a higher-dimensional space where linear relationships suffice. Autoencoders can also be cast in the above framework by parameterizing $h_{ij}(\mathbf{x}_j)$ and $g_i(\cdot)$ with neural networks for hierarchical feature learning. Methods that stress neighborhood preservation, such as t-SNE or UMAP, also fall under $\mathcal{F}$, interpreting $h_{ij}(\mathbf{x}_j)$ to represent graph-based distances or manifold transformations and the incorporation of the same in $g_i(\cdot)$ within the reduced representation.

The choice of PCA and SLP thus serves as a proof of concept: if the framework can encompass such foundational methods from both the linear and non-linear domains, it inherently possesses the generality to accommodate more sophisticated DR approaches. This ensures that $\mathcal{F}$ is not constrained by the specifics of individual algorithms but remains adaptable to the diverse spectrum of techniques used in practice.

# G DISCUSSION AND LIMITATIONS

OptSHAP combines model-specific explanation techniques and the optimization-based strategy to address the challenge of interpreting DRbMs applied to high-dimensional tabular data. Importantly, it is most effective in scenarios where dimensionality reduction techniques achieve significant compression of the original feature space, simplifying the explanation task without incurring significant information loss. In particular, when paired with tree-based models—whose structure aligns well with model-specific explanation methods like TreeSHAP, OptSHAP can deliver explanations that are both computationally efficient and highly faithful.

Conceptually, our framework illustrates how aligning dimensionality reduction with model-specific explanation objectives can create a unified framework that generalizes across both models and preprocessing strategies. Its flexibility lies not just in compatibility with multiple learning components, but in optimizing the interplay between compression and interpretability. This makes it well-suited to domains like healthcare and finance, where high-dimensional data and tree-based models are common and explanations must be.

While OptSHAP offers a novel and scalable approach for explaining DRbMs, its increasing runtime with higher reduced dimensions may limit its use in time-sensitive contexts. Despite this, it remains a promising solution. In Appendix F, we describe how the function class $\mathcal{F}$ can be extended to support nonlinear dimensionality reduction (DR) techniques, enabling broader applicability beyond linear projections. Specifically, we propose approximating nonlinear DR mappings using an interpretable surrogate model, such as a single-layer perceptron, where the approximation's fidelity directly affects explanation quality. We also argue that the predictive model $f$ can be a deep neural network, allowing model-specific methods such as DeepSHAP to generate attributions directly in the reduced-dimensional space. However, when explanations are derived from an approximated mapping, as in the case of nonlinear DR techniques, this introduces additional uncertainty, potentially undermining the reliability of relevance scores when mapped back to the original input space. These observations highlight a broader challenge: how to balance fidelity and interpretability when producing explanations in reduced spaces, especially under nonlinear transformations. Future work should investigate principled ways to quantify and mitigate such uncertainty, enabling more trustworthy attributions across a broader class of DR methods.

While most existing approaches tend to prioritize either stability or complexity, our findings emphasize that effective explainability often hinges on carefully balancing trade-offs, such as stability versus expressiveness, and surrogate fidelity versus interpretability. Future work should explore whether these trade-offs persist across other DR techniques, data distributions, and model architectures. In particular, explicitly formalizing the assumptions underlying surrogate models for nonlinear mappings is crucial for assessing the reliability of resulting explanations. Although our primary focus is on explaining DRbMs, this work also connects to broader interpretability topics, including learned representations and feature attribution in latent or transformed spaces. This can be particularly relevant for applications in medical diagnosis and bioinformatics.

