# OpenReview forum: "OptSHAP: Explaining Dimensionality Reduction-based Models for Tabular Data via Optimization"
_ICLR.cc/2026/Conference — ICLR 2026 Conference Withdrawn Submission_

### Official Review · Reviewer_2hiz · 2025-10-28

**Soundness:** 2
**Presentation:** 3
**Contribution:** 1
**Rating:** 2
**Confidence:** 4

**Summary:**

This paper investigates the feature attribution problem in the context of DRbMs for tabular data, a niche area within explainability research. The authors identify a gap between standard feature attribution methods and DR techniques and propose an optimization-based transformation block $\mathcal{T}$ to bridge this gap. The transformation block strictly follows two fundamental properties and can be used as a plugin for existing explainers to enhance their performance on DRbMs. Finally, the proposed approach is integrated with TreeSHAP and empirically compared with several well-known black-box explainers for validation.

**Strengths:**

- This work is reasonably organized, with a straightforward storyline that facilitates the understanding of the proposed method.
- The authors list fundamental properties of DRbM explainers with appropriate justifications, providing a theoretical foundation that partly supports the design of the method.
- The experimental setting includes validation metrics from complementary perspectives.

**Weaknesses:**

- This paper uses a method name that appears somewhat over-claiming. Although titled “OptSHAP” and leveraging results from TreeSHAP, the proposed method itself has no connection to either SHAP or the Shapley value, according to the description in Section 4. Including “SHAP” in the method name therefore appears misleading.
- The motivation is not convincing:
    - This paper asserts that model-specific methods are unfit for DRbMs. Yet, given the sole focus on linear DR techniques, it is unclear why existing explanation methods would fail to handle such linear transformations. Particularly, such a linear transformation is analogous to a single dense layer without an activation function, which most approaches can accommodate. That said, I agree there are difficulties in processing information flow through a two-stage heterogeneous architecture, but this seems more like an engineering problem rather than a research one.
    - Additionally, linear transformations are generally regarded as explainable/interpretable by themselves. The motivation to develop a specialized approach for such transparent mappings is not well justified. The inherent transparency of linear DRs likely explains why only very few studies specifically investigate their explainability, as they can be covered within a more general framework.
- The method design is not sufficiently motivated. It is promising that the author listed some properties to guide the design; however, it still remains unclear why a $\tanh(\cdot)$, which does not reflect the linear nature of the target transformations, was adopted for the redistribution of attributions.
- The experimental design represents another major weakness. All the other competitors are black-box approaches and relatively old (earlier than 2017). Comparing TreeSHAP, a white-box approach that leverages the advantage in model structural knowledge, with black-box approaches weakens the credibility of the results, making it unclear whether the observed improvements are achieved by the proposed optimization process or from TreeSHAP.

To conclude, this work proposes a plugin for feature attribution in DRbM settings; its contribution and impact appear limited due to insufficient motivation and an unjustified method design.

**Questions:**

- See weaknesses
- Could the authors further clarify the specific design of $\mathcal{T}$? Why is a direct decomposition/redistribution of attributions in the reduced space, according to the proportion of each contribution, insufficient?
- Could the authors clarify the MoRF and ABPC results? Why do many of them show negative values? I am skeptical about the general experimental setup, as negative MoRFs suggest that removing the most relevant features leads to an increase in prediction confidence, which contradicts the expectation.
- How is OptSHAP designed to deliver “variance-reduced” explanations? It appears that stability is guaranteed directly by TreeSHAP, and thus should not be considered a contribution of this work if that is the case.

---

### Official Review · Reviewer_YAFh · 2025-10-31

**Soundness:** 3
**Presentation:** 3
**Contribution:** 2
**Rating:** 4
**Confidence:** 3

**Summary:**

OptSHAP is an optimization-based reallocation method that turns reduced-space attributions into original-feature attributions for DR pipelines.

It does so using a tanh-gated transform keyed to the projection’s Jacobian to preserve signs and approximate efficiency.

It introduces k-Local Stability Score for efficient stability evaluation.

On three high-dimensional tabular datasets across PCA/ICA/LDA and four tree models, OptSHAP improves faithfulness (MoRF/ABPC) and stability (LSS, INFD) compared to KernelSHAP, EIC, and LIME.

Hypothesis tests suggest allocation does not significantly change total importance.

The paper also argues G-DeepSHAP is unsuitable for DRbMs due to denominator blow-ups in latent space distances.

**Strengths:**

The paper addresses a timely problem. The toy example + formal problem statement well motivate “how to get back to features” with sign and efficiency goals.

The tanh-gate transform is conceptually simple, bounded, and sign-aware; is lightweight and avoids sampling variance. The sign-preservation theorem and noise-stability bound are helpful sanity checks (even if conservative)

**Weaknesses:**

The formula is not derived from a theoretical decomposition of SHAP or any formal game-theoretic principle. Instead, it is heuristically designed to satisfy two intuitive desiderata. The paper does not systematically explore alternate formulas or functional forms for the transform T. However, they do justify the use of tanh in the appendix.

OptSHAP do not satisfy the full Shapley axiom set for the original features, nor claim uniqueness of the solution. In such a scenario, I'm not sure if using the term SHAP in the title is a good idea.

**Questions:**

How would tanh compare to Linear mapping, Softmax or ReLU weighting or Normalized loadings?

The current theorems ensure local sign and stability but not closeness to the true Shapley decomposition. Can the authors derive a bound on the deviation of OptSHAP attributions from Shapley-consistent values?

Do results hold if f is logistic regression or a shallow MLP?

---

### Official Review · Reviewer_Xksz · 2025-11-01

**Soundness:** 2
**Presentation:** 2
**Contribution:** 2
**Rating:** 2
**Confidence:** 4

**Summary:**

The work aims particularly to pass the attributions from the reduced feature space to the original, interpretable feature space under a setting of DRbMs. Although SHAP appears to be part of the name, this work does not investigate or propose any approach that facilitates the estimation of Shapley values. Instead, it focuses solely on the redistribution of attributions for the linear DR transformation.

**Strengths:**

-The authors address a rather important topic

**Weaknesses:**

- The references are rather old.
- The approach is very simple and intuitive, as described in section 4, and illustrated by Figure 1. Yet it is confusing why a particular solution should be developed for explaining linear components, which are already explainable/interpretable by themselves.
- There is also a fundamental misunderstanding between the MSE and the efficiency property. The authors refer to IG and SIM-Shapley to justify the slight deviation from the efficiency property. At least for these two articles, the original papers are referring to the MSE of the estimates rather than the deviation in terms of efficiency.

**Questions:**

- Section 4 introduces an optimization problem that is subject to the efficiency property. However, it appears unclear at all why the allocation transform is activated by $\tanh$. Given the linear property of the studied transformations, an attribution redistribution directly following the weights of each feature will lead to a perfect decomposition. For example, $(0.36, 0.3, -0.06)$ and $(0.086, -0.129, 0.343)$, derived from the normalized contribution to each PC, are solutions that strictly align with the linearity of the DRs and adhere to the efficiency property. Why is the proposed method better than this natural solution?

- The experimental results are less convincing, as the allocation method is combined with a much more powerful explainer (TreeSHAP). Shouldn’t the competitors share the same explanation kernel to highlight the benefit of the specific re-allocation design? This is easily doable as all the competitors can be deployed at the reduced feature space and then concatenated by the “Opt”.

---

### Official Review · Reviewer_excZ · 2025-11-05

**Soundness:** 2
**Presentation:** 3
**Contribution:** 2
**Rating:** 4
**Confidence:** 4

**Summary:**

This paper proposes a two-step approach called OptSHAP to compute feature attributions for dimensionality reduction-based models. First, model-specific attribution methods such as TreeSHAP are used to obtain attribution scores for the features in the reduced space. Second, the attribution scores in the reduced space are allocated to the input features by solving an optimization problem.

Theoretically, it is shown that OptSHAP satisfies sign preservation between the reduced and input space and is stable with respect to gradient perturbation. Empirically, OptSHAP (1) outperforms model-agnostic baselines with respect to the insertion/deletion and stability metrics, and (2) satisfies the efficiency property.

**Strengths:**

- This paper addresses the under-explored problem of feature attribution in dimensionality reduction-based models, proposing the desirable property of sign preservation when allocating the reduced-space attributions to input feature attributions.
- The theoretical results are supported by empirical results on simulation or real-world data.
- OptSHAP outperforms model-agnostic baselines in both the insertion/deletion metrics and stability metrics.

**Weaknesses:**

- The empirical results focus entirely on tree-based models as the predictor. Although the authors motivate this choice in Remark 4.3, the limited scope still raises the question about whether OptSHAP works **empirically** for other predictor classes such as neural networks.
- Theorem 4.4 is motivated by approximation errors that can happen in gradient computation. It is unclear whether Gaussian nosies are reasonable distributions for modeling such approximation errors. The choice of Gaussian noises in Theorem 4.4 need to be better connected to the problem of approximation errors.
- Lines 394-396 mention that individual features may contribute positively or negatively to the output, and the implication is that attribution values should be balanced around zero. It is unclear why the implication holds. For example, it's possible that more samples have negative contribution from a given feature. In this case the beeswarm plot in Figure 6 can look skewed (such as the LIME example).

**Questions:**

- Does OptSHAP outperform model-agnostic baselines when the downstream predictor is not tree-based, such that attribution methods other than TreeSHAP need to be used? Overall, is the empirical success of OptSHAP restricted to tree-based predictors?
- Why are gradient approximation errors modeled by Gaussian nosies?
- How is the second expectation in Section 6.4 justified theoretically? Do we expect the second expectation to hold for all datasets?

---

### Note · Authors · 2025-11-15

I have read and agree with the venue's withdrawal policy on behalf of myself and my co-authors.